# Can rhythm-induced attention improve the perceptual representation?

**Asaf Elbaz**  *, Yaffa Yeshurun

Department of Psychology & Institute of Information Processing and Decision Making, University of Haifa, Mount Carmel, Haifa, Israel

* asafelbaz@gmail.com

## Abstract

Temporal attention can be entrained exogenously to rhythms. Indeed, faster and more accurate responses were previously found when the target appeared in-phase with a preceding rhythm in comparison to when it was out of phase. However, the nature of this rhythm-induced attentional effect is not well understood. To better understand the processes underlying rhythm-induced attention, we employed a continuous measure of perceived orientation and a mixture-model analysis. A trial in our study started with a sequence of auditory beeps separated by a fixed inter-beeps interval in the regular (rhythmic) condition or by variable inter-beeps intervals in the irregular condition. A visual target–a line embedded in a circle–followed the sequence. The 'critical' interval between the last beep and the target was chosen randomly from several possible Inter-Onset Intervals (IOIs), of which only one was in-phase with the rhythm. The target was followed by a probe line, and the participants were asked to rotate it to reproduce the target's orientation. The measure of performance for a given trial was the difference in degrees between the orientation of the target and that reproduced by the observer. We found that guessing rate was lower with regular than irregular rhythms. However, there was no effect of rhythm type (regular vs irregular) on the quality of representation (measured as the variability in reproducing the target). Furthermore, the rhythm effect was present only when rhythm type was fixed within a block, and it was found with all IOIs, not just the in-phase IOI. This lack of specificity suggests that these results reflect a general effect of rhythm on alertness.

## Introduction

As with spatial attention, temporal attention can be endogenous or exogenous [e.g., 1; 2; 3; 4]. Several studies demonstrated that rhythms can entrain exogenous temporal attention [e.g., 1; 5; 6; 7, for a recent review see 8]. Typically, a rhythm with a fixed inter-onset interval (IOI, the interval between the onset of one stimulus to the onset of the following stimulus) precedes target presentation, and the critical IOI (the last IOI prior to target onset) could have one of several durations. When the critical IOI matches the preceding IOIs of the rhythm (i.e., it is in-phase with the rhythm), performance improves in terms of accuracy and/or response time (RT), in comparison to a too short or too long critical IOI (i.e., IOIs that are out of phase with the rhythm). Rhythm effects were found for various tasks including pitch-judgment task [1],

**Data Availability Statement:** The data are held in a public repository at DOI: 10.17605/OSF.IO/8CWFZ.

**Funding:** This study was supported by the Israel Science Foundation Grant 1081/13 to YY https://www.isf.org.il/ The funders had no role in study

design, data collection and analysis, decision to publish, or preparation of the manuscript.

**Competing interests:** The authors have declared that no competing interests exist.

detection task [3; 5; 7], and perceptual discrimination [9]. Critically, these effects were found even when the rhythm did not predict target onset (i.e., target onset could equally likely follow all possible critical IOIs). These results suggest that these rhythm effects reflect exogenous attention allocation to the specific point in time that matches the rhythm's IOI [e.g., 1; 5; 10; 11; 12]. Based on such rhythm effects, the dynamic attending theory (DAT) suggests that the behavioral entrainment to a rhythm is due to entrainment of internal oscillators to external rhythms. This then leads to a narrower attentional focus with regular (isochronous) rhythms than with irregular (asynchronous) rhythms [1]. In this study, we examined which mechanisms underlie the typical performance improvement that is observed when attention is entrained by a rhythm.

To that end, we employed a continuous-report task in which participants are asked to reproduce the orientation of the target by adjusting a probe's orientation. Thus far, only discrete-report tasks were used to examine rhythm-induced attention. With these tasks, the observer is typically required to decide between two alternative responses (e.g., indicate whether the target's orientation was upright or inverted). Unlike discrete-report tasks, with a continuous-report task, the observer is typically asked to reproduce one of the target's features as close as possible on a continuous scale. For instance, the observer is asked to rotate a probe line to assume an orientation that is as close as possible to the target's orientation (Fig 1). In this case, the measure of performance for a given trial is the difference in degrees between the orientation of the target and that reproduced by the observer (e.g., if the target's orientation was 60˚ and the observer rotated the probe to an orientation of 80˚, the error measured in this trial is +20˚). Although the two task types are closely linked, measurements of continuous response reflect a continuum of answers on a specific scale that is sensitive to a behavior that is driven by a less-than-optimal target representation, because one has the option to generate a partial (estimated) answer rather than choose between limited options. Moreover, combining

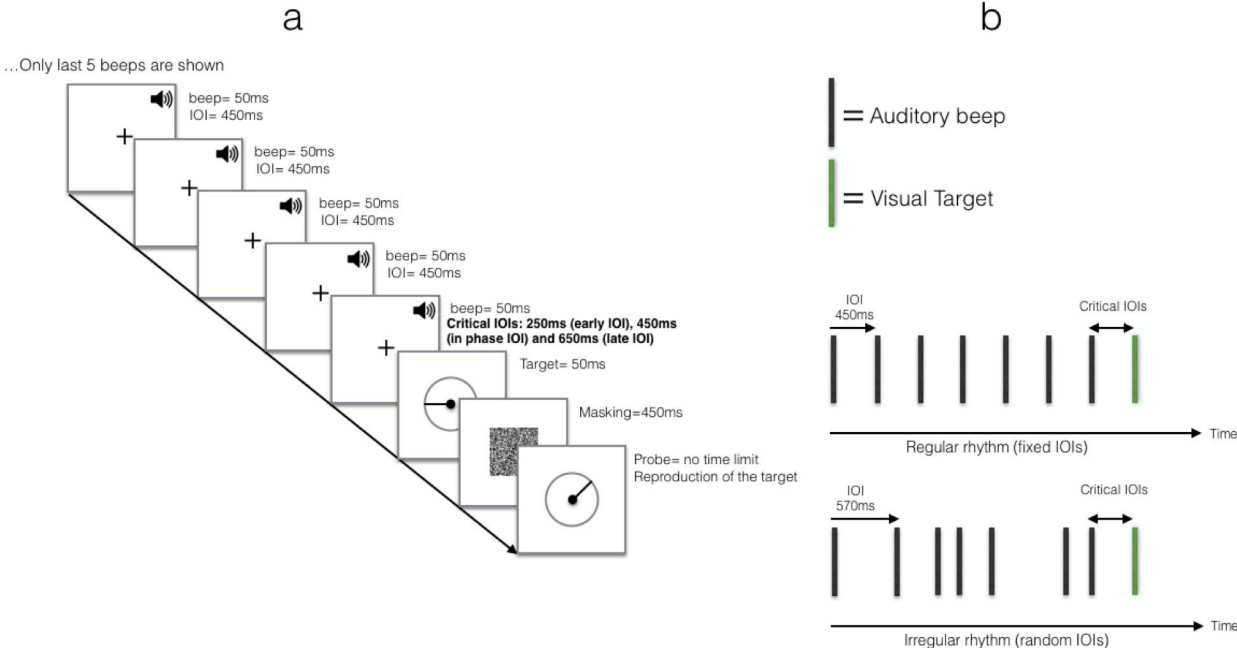

**Fig 1.** (a) A schematic example of a trial in Experiment 1. A rhythm (in this example it is a regular rhythm with a fixed IOI) precedes the visual target, which appears after one of 3 critical IOIs (early IOI-250ms, in-phase IOI- 450ms, or late IOI- 650ms). (b) A schematic description of the different rhythm types.

this task with a mixture model, as detailed below, allows us to better understand the processes underling the observed effect, rather than merely report its existence.

The most commonly used model to analyze continuous-report data is the mixture model [e.g., 13; 14; 15]. It was originally used in the context of visual short-term memory [e.g., 15], but it is also useful for studies of visual perception [e.g., 16; 17; 18; 19; 20]. According to the standard mixture model, the overall error distribution reflects the combination of two distributions: A Gaussian distribution and a uniform distribution. The Gaussian distribution is centered around the orientation of the target (i.e., error = 0), and it is the distribution of errors that are the result of less than perfect target representation. The uniform distribution is a distribution of errors that are the result of pure guessing (i.e., when the reproduction of all line orientations is equally probable). The combination of these two distributions generates a mixture distribution with three parameters: (1) the width of the Gaussian distribution (*SD*). This parameter reflects the error variance of trials in which the target was at least partially perceived. It conveys the precision of the encoding process, or the precision of the representation [e.g., 16; 20]; the smaller the *SD* the higher the encoding precision. (2) The height of the uniform distribution (*g*). This parameter indicates the guessing rate (i.e., it reflects the proportion of trials, out of the total number of trials, in which the participant provided a random response. For example, if *g* = 0.2 this means that on 20% of the trials the error is due to guessing). The larger the *g* the higher the guessing rate [e.g., 16; 20]. Thus, from now on we refer to the *g* parameter as the guessing rate. (3) The mean of the Gaussian distribution (*μ*). This parameter reflects potential biases. If the Gaussian distribution of errors is indeed centered around 0 (*μ* = 0) this suggests that there is no bias. Because our experiments included no source for bias, nor did we find any evidence that indicates a bias in target reproduction (see Results section), we did not include this factor in our final analysis, thereby reducing the number of free parameters. The model we used is summarized with the following equation that includes two free parameters [15]:

$$p(\theta) = (1 - g)f(\theta)\sigma + g/360 \tag{1}$$

where *θ* is the value of the estimation error, *g* is the proportion of trials in which participants are randomly guessing, *f(θ)σ* is the von Mises distribution (the circular analogue of the Gaussian distribution) with mean of zero and *σ* standard deviation (*SD*).

To take advantage of the continuous-report task to gain better understanding of rhythm-induced attention, we combined this task with auditory rhythms. We chose to employ auditory rather than visual rhythms because previous studies found stronger entraining qualities for auditory rhythms over visual rhythms [e.g., 21; 22]. Critically, previous studies also showed that rhythm-induced attentional effects are not limited to a unimodal design [e.g., 23; 24; 25]. For example, in Miller, Carlson, and McAuley's study [26], a visual target followed an auditory rhythm, and was either in-phase or out of phase with the rhythm. They found that saccade latency to the visual target was reduced and discrimination accuracy increased when target onset was in-phase with the preceding auditory rhythm. Thus, at the beginning of each trial in our study, the participants heard an auditory sequence of beeps that was either regular (i.e., the beeps were separated by a fixed IOI–an isochronous rhythm) or irregular (i.e., the beeps were separated by randomly varying IOIs–an asynchronous rhythm). The rhythm was followed by a briefly presented target, which was a line with a random orientation presented inside a circle. Importantly, the target could appear after several possible critical IOIs, of which only one was in-phase with the preceding rhythm. The participants were then asked to rotate a probe in order to reproduce the target's orientation (Fig 1). We measured the error score as the difference between the actual orientation of the target and the orientation generated by the

participant. This error score ranged between 0˚ (perfect reproduction) and ±180˚. Analyzing the error data with the mixture model allowed us to test whether rhythm entrainment can affect the quality of the perceptual representation of the target (*SD*), the guessing rate (*g*), or both. Thus, if the regular rhythm can indeed entrain temporal attention, thereby affecting perceptual processing, we should find higher quality of representation and/or lower guessing rate when the target is in-phase with the regular rhythm in comparison to trials in which the target is out of phase or trials in which the target follows an irregular rhythm. Thus, by using a continuous-report task and mixture-model analysis, we can gain better understanding of how rhythms may shape our perception.

## Experiment 1

### Method

**Participants.**   A total of 23 students from the University of Haifa participated in the experiment. Three participants were excluded from the final analysis because with their data the modeling procedure failed to converge (i.e., the algorithm could not find parameters that produced a good fit, likely due to too noisy data, and because these parameters are the dependent variables on which the analyses are conducted, we had to exclude these participants). One participant was excluded because her guessing rate was above 50%. Thus, the final analysis was performed on 19 participants. All participants had normal or corrected to normal vision, normal audition, and no history of neurological or psychiatric disorder. All participants were naive to the purpose of the study. We chose this number of participants based on the average number of participants in studies of rhythm entrainment [e.g., 23; 27]. Furthermore, we calculated the sample size required in order to observe a significant rhythm effect. We conducted a power analysis with G∗Power [28], using an alpha of 0.05, power of 0.95, and the effect size found in two other rhythm studies [5, dz = 0.51; 7, dz = 0.61]. We found that the minimum sample size required is fourteen and sixteen, respectively, and because our study involved a continuous measurement a slightly higher sample size seemed desirable. This study adhered to the Declaration of Helsinki. All experiments were approved by the ethics committee of the University of Haifa (293/15). All observers signed a consent form.

**Stimuli, apparatus.**   The experiment was conducted in a dimmed room. The beeps (500 Hz, 60dB) were presented via an Over-Ear headphone. Visual stimuli were presented at the center of a 17-in CRT screen (ViewSonic G75f, with 100 Hz refresh rate) on a gray background (RGB = 128 128 128, viewing distance = 57 cm). The visual target was a line (1˚ of visual angle) with a random orientation presented within a circle (radius = 1˚). The target's luminance was individually adjusted with a staircase procedure during the practice phase to allow ~80% accuracy (range: RGB 46 46 46—RGB 120 120 120). The mask was a static random-dots square (2.6˚). The probe was similar to the target but with a different luminance [RGB 0 0 0] and a randomly chosen orientation. Stimuli presentation and response acquisition were handled using the Psychophysics toolbox [29] for MATLAB (version 7.5.0, Mathworks, Natick, Massachusetts).

**Procedure.**   At the beginning of each trial, the participants watched a fixation mark at the center of the screen and heard a pattern of seven identical beeps, each presented for 50ms, in a regular or irregular rhythmic structure, mixed randomly within a block. In the regular rhythm, the beeps were separated by a fixed IOI of 450ms. In the irregular rhythm, the following IOIs–200ms, 230ms, 450ms, 500ms, 570ms, 750ms–were randomly permutated on each trial (total duration equals that of the regular rhythm). The rhythm was followed by a briefly (50ms) presented target that replaced the fixation mark. Critically, the target appeared after one of the following critical IOIs with equal probability– 250ms, 450ms, 650ms. The critical IOI of 450ms

was in-phase with the regular rhythm, and the other critical IOIs were out of phase (too early - 250ms or too late - 650ms). The target was followed by a mask that lasted for 400ms, and it was replaced by the probe. The participants were instructed to rotate the probe (by pressing the left and right arrow keys) to reproduce the target's orientation as close as possible. No additional instructions were given regarding the preceding rhythm to ensure that any attentional effects that may emerge will be purely exogenous. When satisfied with their decision, participants pressed the upper arrow key and the next trial started. The experiment lasted for approximately 90 minutes and included 450 trials: 120 trials for each critical IOI + 90 (20%) catch trials in which no target was presented to minimize the 'foreperiod' effect–improved performance with longer critical IOIs [e.g., 7]. This effect is common when different critical IOIs are mixed within a block, and it is presumably due to the fact that expectancy builds up as time elapses [e.g., 30]. Participants pressed the letter "N" to report that no target appeared. They were allowed to take a short break every 108 trials. After each block, participants received feedback about their performance describing the percentage of trials in the preceding block, in which precision was 'high' (defined as orientation reproduction with a lower than ±10˚ difference from the target's orientation).

## Results and discussion

We used the Memtoolbox [31] to fit each observer's responses with the standard mixture model that includes 2 parameters–the *SD* of the Gaussian distribution and the height of the uniform distribution (*g*) as detailed in Eq 1. We chose the model with 2 parameters because there was no theoretical reason to expect a consistent bias. We nevertheless used 3 criteria (Akaike Information Criterion—AIC, corrected Akaike Information Criterion—AICc, Bayesian Information Criterion—BIC) to test which model fits the data better, the standard mixture model with bias, which includes 3 parameters, or the standard mixture model without bias, which includes only 2 parameters. All 3 tests favored the model without bias. The fit of the model, with the 2 parameters, to the data can be seen in Fig 2. We then extracted these parameters for each participant and analyzed them using a repeated-measures two-way (rhythm type, critical IOI) analysis of variance (ANOVA). These analyses revealed a marginally significant main effect of critical IOI on guessing rate (F(2,36) = 3.26, p = 0.0501, $\eta_p^2$ = 0.15); guessing rate was lower with longer critical IOIs (Fig 3). As mentioned above, this 'foreperiod effect' is common when different critical IOIs are mixed within a block, and it is presumably due to the fact that expectancy builds up as time elapses [e.g., 30]. We attempted to minimize this effect by incorporating catch trials [e.g., 7], but it was, nevertheless, present in our study. All other effects (with guessing rate or *SD*) did not reach statistical significance.

The lack of effects that involve the rhythm manipulation, and particularly the lack of a significant rhythm x IOI interaction did not match our expectations, nor do these findings match previous studies demonstrating rhythm effects [e.g., 4; 5; 7]. Perhaps these results are due to the mixed design employed here, in which rhythm type varied randomly within a block. Indeed, many of the studies who found involuntary attentional entrainment to rhythms used a blocked design [e.g., 5; 32; 33; 34], and we found in a recent study [35] that a blocked design was required for the emergence of a rhythm effect. This possibility is tested in Experiment 2.

## Experiment 2

### Method

**Participant.**    A total of 18 students from the University of Haifa performed the experiment. One participant was excluded because her guessing rate was above 50%. The final analysis was performed on 17 participants. All participants had normal or corrected to normal

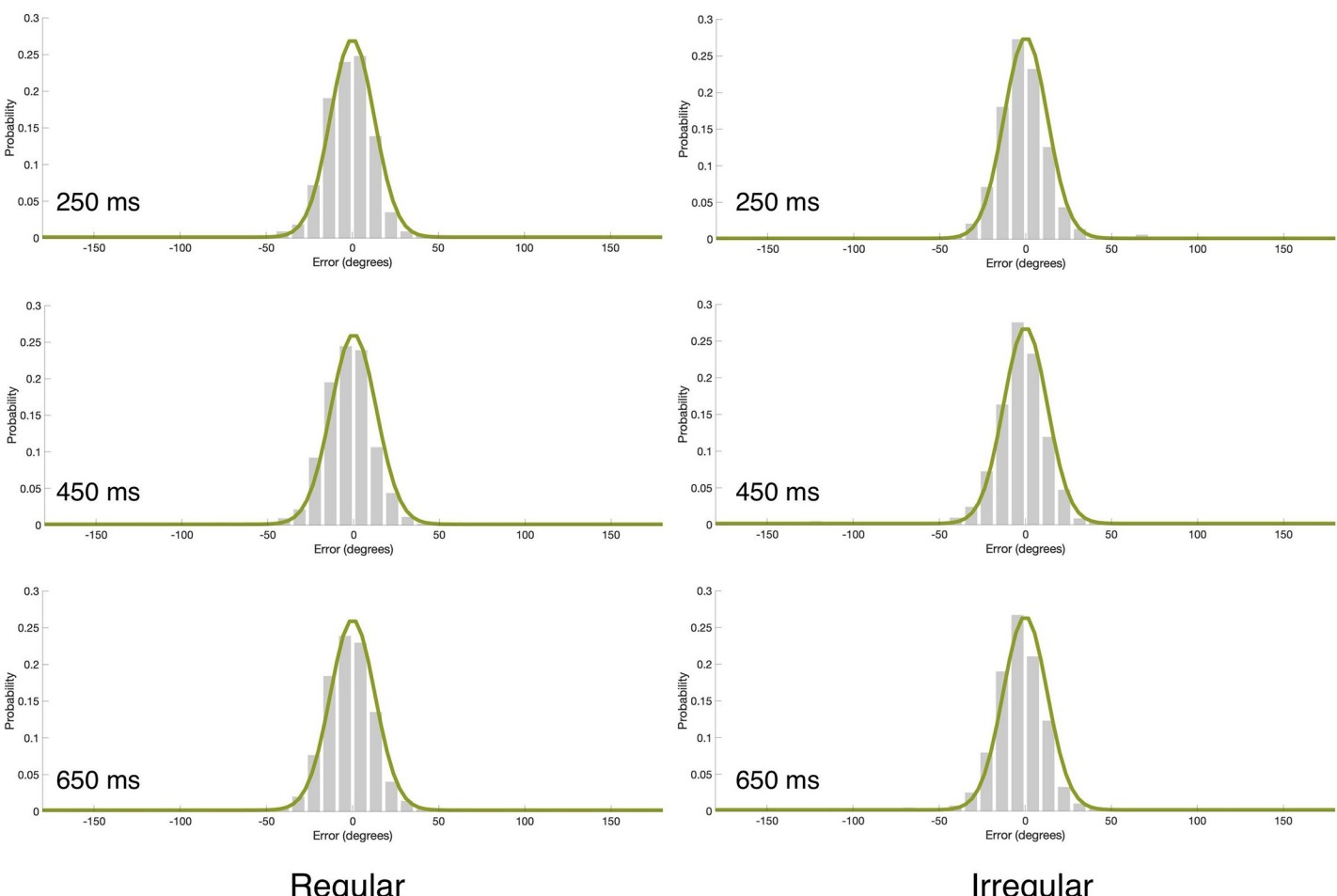

**Fig 2. Mean error distributions (gray bars) and mixture model fits (green line) as a function of rhythm type and critical IOI in Experiment 1.** Each panel corresponds to a different critical IOI in the regular and irregular rhythm conditions. Model fits were generated by using the model's parameters averaged across participants. These histograms were generated for visualization purpose only; the statistical analyses were performed based on fitting the model to individual data.

vision, normal audition, no history of neurological or psychiatric disorder, and all were naive to the purpose of the study.

**Stimuli, apparatus and procedure.** This experiment was similar to Experiment 1 with the following changes. We employed a blocked design that included 4 blocks (120 trials per block of which 25% were catch trials). We also used different IOIs that are closer to the average spontaneous tapping rate, thus potentially making entrainment easier. For example, Hove et al. [36] found tapping synchronization was more stable with the slow tempo (600ms) than with the fast tempo (400ms). In the regular rhythm blocks, the beeps were separated by a fixed 650ms IOI. In the irregular rhythm blocks, we used a random permutation of the following IOIs: 100ms, 300ms, 500ms, 800ms, 900ms, 1300ms (total duration equaled that of the regular rhythm). We also changed the critical IOIs to 250ms (too early), 650ms (in-phase) and 1050ms (too late). All critical IOIs in both conditions were presented with equal probability and mixed within blocks. We increased the differences between the different critical IOIs because with larger differences it might be easier to observe attentional benefits that are due to rhythm entrainment. We also increased the amount of catch trials (25% instead of 20%, as in Experiment 1), in an attempt to minimize the foreperiod effect. Thus, overall the experiment

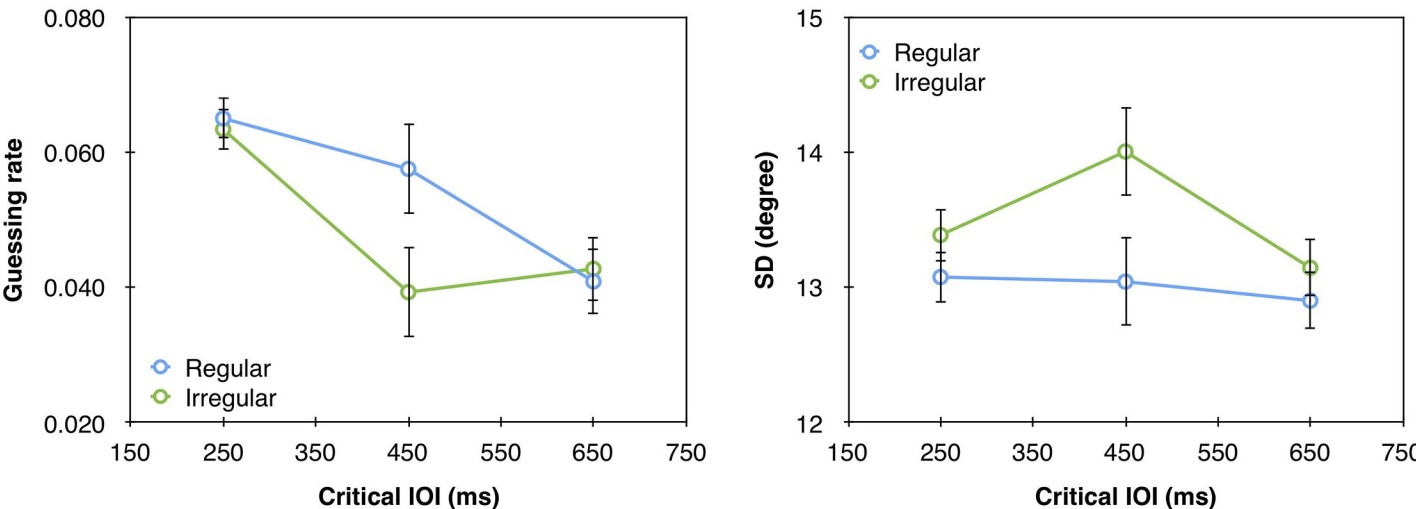

**Fig 3.** (a) Guessing rate (*g*) and (b) *SD* as a function of the different critical IOIs in the regular and irregular rhythm conditions in Experiment 1. The in-phase critical IOI is 450ms. Error bars represent 1 Standard Error of the Mean (SEM).

included 480 trials:120 for each critical IOI + 120 catch trials. Finally, we presented an additional 8[th] beep simultaneously with the target. We assumed that by increasing the number of beeps and linking the target to the contextual rhythm, the entrainment to the rhythm may better manifest itself. Block order was counterbalanced across participants.

## Results and discussion

The analyses were similar to Experiment 1. The fit of the model to the data is presented in Fig 4. These analyses revealed a significant main effect of critical IOI on guessing rate (F(2,32) = 5.57, p = 0.0084, $\eta_p^2$ = 0.26) but not on *SD* (F = 2.02, p = 0.1499). As in Experiment 1, the guessing rate was lower for longer IOIs (Fig 5). Thus, increasing the percentage of catch trials to 25% did not eliminate the foreperiod effect. Importantly, when the type of rhythm was fixed within a block, we found a significant main effect of rhythm type with both guessing rate (F(1,16) = 8.22, p = 0.0112, $\eta_p^2$ = 0.34), and *SD* (F(1,16) = 5.075, p = 0.0388 $\eta_p^2$ = 0.24). Specifically, lower guessing rate and higher representation quality (lower *SD*) were found for targets that appeared after a regular than irregular rhythms. This general effect of rhythm across critical IOIs likely reflects general increase in alertness, as discussed in more details in the General Discussion section.

The critical IOI x rhythm type interaction was not significant with both guessing rate (F<1) and *SD* (F<1). Thus, employing a blocked design did not provide evidence for a specific attentional allocation to the point in time that was in-phase with the rhythm. Could this lack of a specific effect stem from the fact that the in-phase critical IOI was also the average critical IOI? Perhaps the participants developed some temporal expectations regarding this average regardless of rhythm type. It is widely accepted that the ability to extract statistical regularities from the sensory input is a fundamental cognitive ability [e.g., 37–41]. Thus, it is possible that our participants extracted the mean IOI across all trials, and accordingly developed an expectation that the target will follow this averaged IOI. Alternatively, the current results reflect a mixture of the rhythm entrainment and foreperiod effects that is hard to disentangle with 3 values of critical IOI. In Experiment 3 we avoid these obstacles by employing 4 critical IOIs and ensuring that the in-phase critical IOI is different from the average critical IOI.

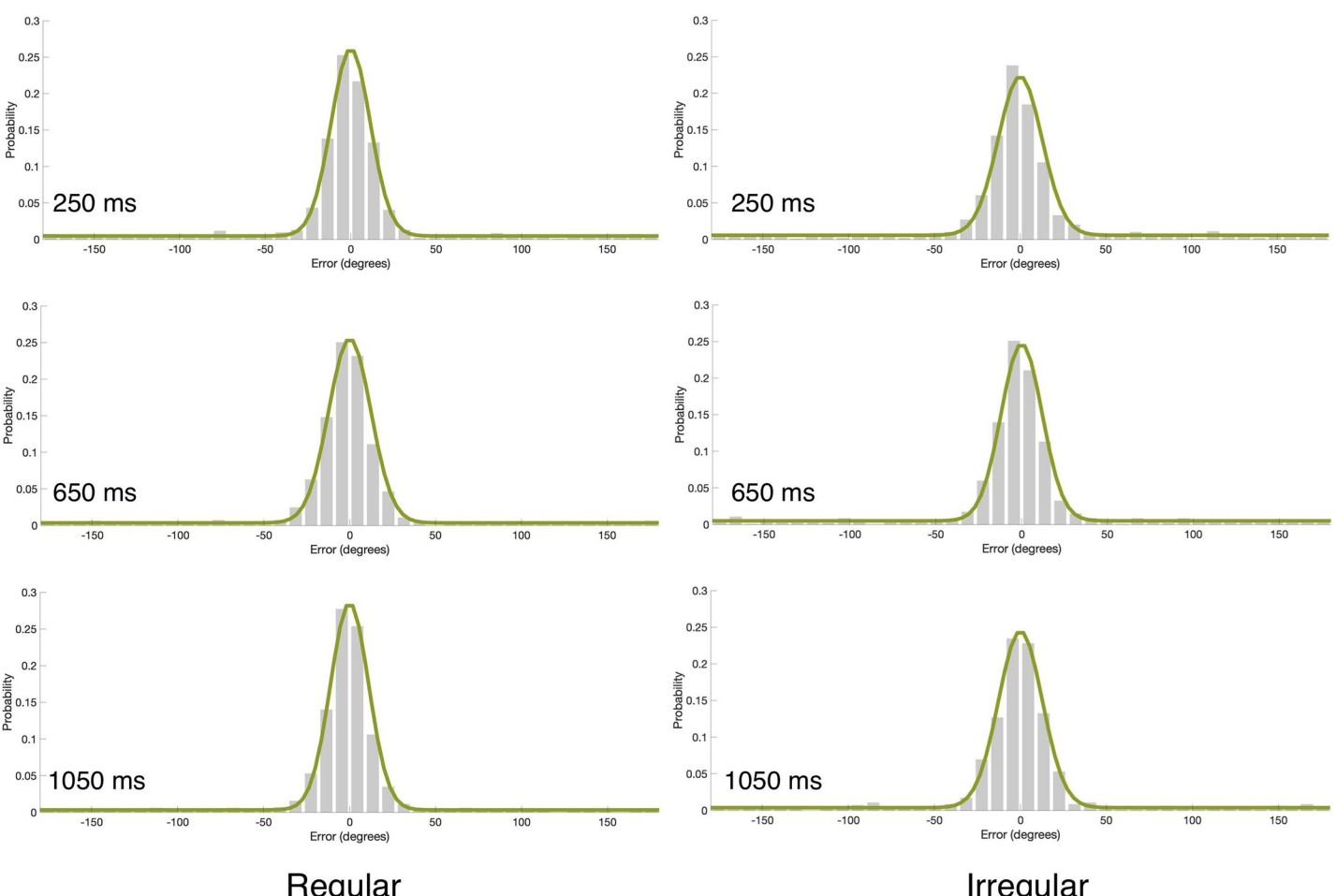

**Fig 4. Mean error distributions (gray bars) and mixture model fits (green line) as a function of rhythm type and critical IOI in Experiment 2.** Each panel corresponds to a different critical IOI value in the regular and irregular rhythm conditions. Model fits were generated by using the model's parameters averaged across participants. These histograms were generated for visualization purpose only; the statistical analyses were performed based on fitting the model to individual data.

## Experiment 3

### Method

**Participants.**   A total of 25 students from the University of Haifa performed the experiment. Five participants were excluded due to model failure to converge. The final analysis included a total of 20 participants. All participants had normal or corrected to normal vision, normal audition, no history of neurological or psychiatric disorder, and all were naive to the purpose of the study.

**Stimuli, apparatus and procedure.**   This experiment was similar to Experiment 2 (including a blocked design) except for the following changes. In this experiment all trials included a target. This change was introduced because we found in Experiments 1 and 2 that the catch trials did not eliminate the foreperiod effect, and because we wanted to include 4 critical IOIs without having to reduce the number of trials per critical IOI. Instead of the catch trials, we relied on the presence of the irregular condition as a control for the foreperiod effect [6]. The critical IOIs were: 200ms (too early), 650ms (in-phase), 950ms (too late) and 2050ms (too late). The irregular rhythm blocks included a random permutation of the following IOIs:

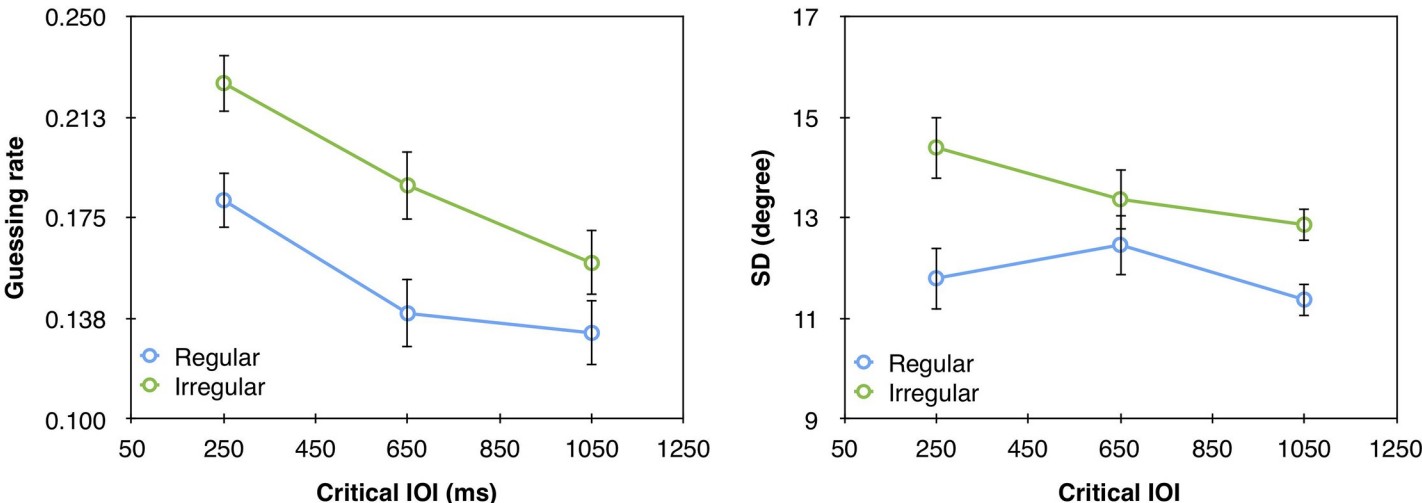

**Fig 5.** (a) Guessing rate (*g*) and (b) *SD* as a function of the different critical IOIs in the regular and irregular conditions in Experiment 2. The in-phase IOI is 650ms. Error bars represent 1 SEM.

100ms, 300ms, 500ms, 800ms, 900ms, 1300ms (total duration equals that of the regular rhythm). Overall the experiment included 480 trials– 120 trials for each critical IOI.

## Results and discussion

The statistical analyses were similar to Experiments 1 and 2. The fit of the model to the data can be seen in Fig 6. These analyses revealed a significant main effect of critical IOI on guessing rate ($F(3,57) = 19.858$, $p<0.0001$, $\eta_p^2 = 0.51$) but not on *SD* ($F<1$). As was found for Experiments 1 and 2, guessing rate was lower with longer critical IOIs (Fig 7). This foreperiod effect was particularly large (effect size: $\eta_p^2 = 0.15, 0.26, 0.51$ in Experiments 1–3 respectively), which is expected given that in this experiment there were no catch trials. Also similar to Experiment 2, we found a main effect of rhythm type on guessing rate ($F(1,19) = 18.04$, $p = 0.0004$, $\eta_p^2 = 0.49$), but not on *SD* ($F<1$). Specifically, when the targets appeared after a regular rhythm the guessing rate was lower than when they appeared after an irregular rhythm. As mentioned above, we believe this general effect of rhythm reflects general increase in alertness, and we further discuss this in the General Discussion section.

The IOI x rhythm interaction was not significant with guessing rate ($F = 1.5$, $p = 0.218$, $\eta_p^2 = 0.07$), suggesting that the regular rhythm did not have a specific effect on guessing rate, but the interaction was significant with *SD* ($F(3,57) = 4.07$, $p = 0.011$, $\eta_p^2 = 0.18$). However, the pattern of this interaction is different from the one expected given previous demonstrations of rhythm-induced attention [e.g., 4; 5; 7]. That is, if the regular rhythm entrains attention to the specific points in time that match the rhythm, the quality of representation should be higher (*SD* should be smaller) for targets that follow the in-phase critical IOI with the regular rhythm in comparison to the other critical IOIs of this rhythm condition and particularly in comparison to the corresponding critical IOI (650ms) of the irregular rhythm. Instead, we found that the quality of representation was higher (*SD* was smaller) in the regular than irregular rhythm condition with the longest critical IOI, but was lower for the shortest critical IOI. Moreover, post-hoc analyses with Bonferroni correction showed that the only pairwise comparison that reached statistical significance was the difference in *SD* between the shortest and longest critical IOIs of the regular rhythm ($p = 0.04$). Thus, clearly, this pattern of interaction does not reflect a specific attentional allocation to the in-phase point in time.

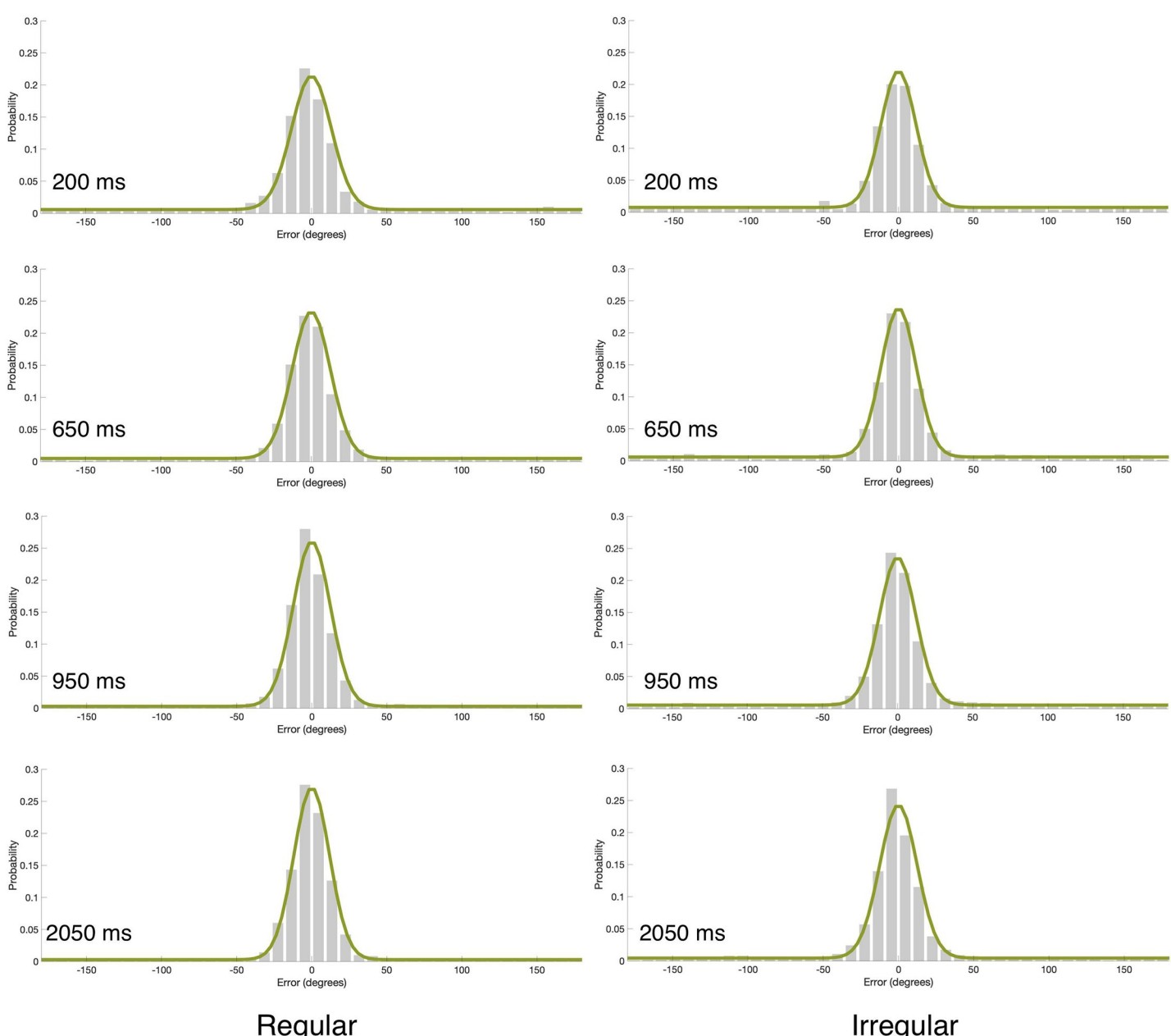

**Fig 6. Mean error distributions (gray bars) and mixture model fits (green line) as a function of rhythm type and critical IOI in Experiment 3.** Each panel corresponds to a different critical IOI value in the regular and irregular rhythm conditions. Model fits were generated by using the model's parameters averaged across participants. These histograms were generated for visualization purpose only; the statistical analyses were performed based on fitting the model to individual data.

## General discussion

In this study, we explored the processes underlying rhythm-induced exogenous temporal attention. In all experiments, target presentation was preceded by a sequence of auditory beeps separated by a fixed IOI in the regular rhythm condition or by variable IOIs in the irregular condition. Importantly, the 'critical' interval between the last beep and the target was chosen randomly from several possible IOIs, of which only one was in-phase with the regular rhythm. The target was followed by a probe, and the participants were asked to rotate it to reproduce

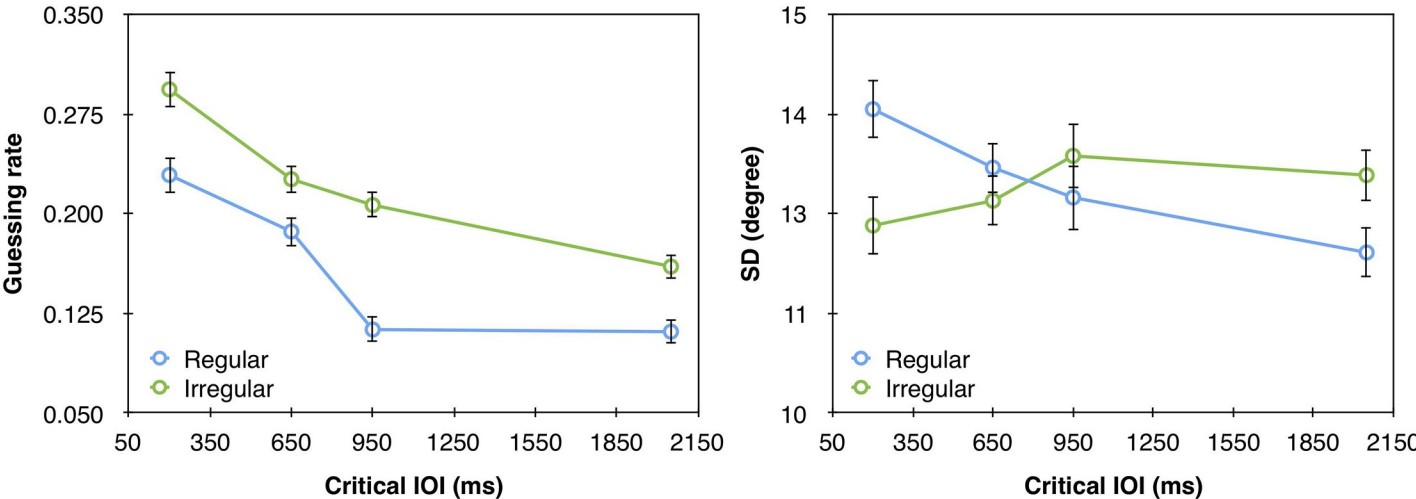

**Fig 7.** (a) Guessing rate (*g*) and (b) *SD* as a function of the critical IOIs in the regular and irregular conditions of Experiment 3. The in-phase IOI is 650ms. Error bars represent 1 SEM.

the target's orientation. Using a mixture-model analysis, we examined whether rhythm-induced attention influenced the quality of the target's representation and/or the guessing rate. In Experiment 1 rhythm type varied within a block, and no effect of attention emerged. In contrast, in Experiments 2 and 3, rhythm type was fixed within a block, and here rhythm effects did emerge. This finding suggests that a blocked designed is preferable for the emergence of rhythm-induced attention, and it is consistent with the fact that most of the studies that demonstrated rhythm-induced attentional effects indeed used a blocked design [e.g., 4; 5; 35]. Perhaps with a mixed design there is a high degree of uncertainty regarding the temporal structure of a trial, and this obscures any effects that are related to this temporal structure. The rhythm effects that were found in Experiments 2 and 3, were mostly manifested as a general reduction in guessing rate. That is, in both experiments, trials on which the target appeared after the regular rhythm, led to lower guessing rate in comparison to the irregular rhythm. Yet, this effect was found with all critical IOIs, regardless of whether or not they were in-phase with the rhythm. This general effect of rhythm likely reflects increased alertness [e.g., 42]. That is, the regular rhythm might have induced automatic arousal increase, which can occur independently of temporal expectations regarding a specific point in time [for a discussion of temporal orienting vs. alertness see 43]. For example, Hackley et al. [44] found that alerting cutaneous stimuli reduced RTs even when participants knew in advance exactly when the task-relevant visual stimulus would appear. This finding suggests that the alerting rhythm generated a bottom-up alertness increase that is different from the top-down temporal expectations. The two mechanisms are also mediated by different brain areas [44]. Phasic arousal seems to reduce the threshold for response selection within a circuit involving the supramarginal gyrus. Temporal expectancy, on the other hand, was mediated by the executive control areas, as well as the right frontal pole and the left middle temporal gyrus. In spatial attention, Matthias et al. [45] also found that phasic alerting could shift spatial distribution of attentional weighting and increase processing speed. In another study, phasic alertness was linked to increased conscious perception as a response to an auditory alerting cue, both objectively and subjectively [46]. Finally, in a recent study conducted in our lab, we found very similar results [35]. In that study, we have examined whether a familiar rhythm can serve as a hybrid cue (top-down–bottom-up) for temporal attention. Target presentation was preceded by a non-predictive familiar, regular or

irregular rhythm, and the participants performed a 2FAC orientation discrimination task. We found decreased RT in the familiar rhythm in comparison to the irregular rhythm, which was particularly pronounced with the critical IOI that was in-phase with the familiar rhythm. This finding suggests that familiar rhythms can direct attention to a specific point in time that matches the previously learned temporal structure of the rhythm. However, like the current study, we only found a general effect with the regular rhythm; the participants of that study responded faster to targets that appeared after the regular rather than irregular rhythms across all IOIs. Thus, this general effect of rhythm seems to be a robust effect.

The general performance improvement found with the regular rhythm may also be related to the degree of temporal uncertainty involved in the rhythm. As Lawrence and Klein [47] suggested, exogenous temporal attention should be studied in the absence of any contingency between the target and the cue. Yet, typical rhythm studies are contingent by nature, because often the rhythm precedes the target [e.g., 3; 5; 35]. Given such contingency, it is possible that with the regular rhythm, the participants can better estimate the time of the offset of the last rhythmic cue which signifies the upcoming target onset. That is, participants may conceptualize the preceding rhythm and target onset as two separate temporal events, each with its own temporal uncertainty. The less uncertainty each event posits, the easier it is to segregate these two temporal events. In other words, although the rhythm did not predict the exact time of target onset, it enhanced the ability to temporally segregate the onset of the target from its preceding rhythm by increasing the ability of the participants to predict the time of target onset. Because the regular rhythm inherently involves less temporal uncertainty than the irregular rhythm, it might have allowed the participants to better estimate when the second temporal event (i.e., target onset) will occur. Importantly, because the current study employed a continuous-report task and a mixture-model analysis we could examined the mechanisms that underlie this general effect. Specifically, because general rhythm effect on the *SD* parameter were not consistent (were only found in Experiment 2), we cannot provide evidence in support of a mechanism that improves target representation. Instead, we found a robust general increase in the *g* parameter or the guessing rate, and because the guessing rate indicates the rate at which the target was not registered at all, Agaoglu et al. [16] suggested that this parameter reflects the signal-to-noise ratio (SNR). We, therefore, can conclude that this general effect of rhythm is mediated by increased SNR.

Unlike the general effect of rhythm that was found in both experiments, we did not find, in any experiment, evidence for a specific attentional allocation to the point in time that was in-phase with the rhythm. In none of the experiments a rhythm x IOI interaction emerged for the guessing rate measurement, and although this interaction was significant for the quality of representation measurement (Experiment 3), the interaction pattern did not follow the pattern expected given an attentional allocation to the in-phase point in time. The lack of a specific advantage for the in-phase point in time with the regular rhythm was surprising given previous reports of such specific effects [e.g., 1; 3; 5]. One may wonder whether the lack of a specific advantage was due to the cross-modal design we employed in this study (i.e., an auditory rhythm coupled with a visual target). However, as we indicated before, several previous studies have already demonstrated effects of rhythm-induced attention with a cross-modal design [e.g., 23; 24; 25; 26]. Still, such attentional effects may be less robust under cross-modal setting. That is, when the rhythm and the task-relevant target belong to different modalities, the emergence of rhythm induced effects may be more susceptible to methodological modifications [48]. A failure to replicate the specific performance enhancement for the point in time that is in-phase with an isochronous rhythm was recently reported by Bauer et al. [49; see also 50], and as described above was also the case with a recent study we performed on isochronous and familiar rhythms [35]. Additionally, no evidence was found for reduced attention blink

when the onset of the 'blinked' target matched the rhythm [51], nor did the presentation of a pseudoword near a rhythmic peak improved its later recognition [52]. Furthermore, some studies, which reported a specific rhythm-induced effect, used only one critical IOI–the in-phase critical IOI–and compared it to an irregular rhythm [e.g., 9; 27]. For example, in the study by Cutanda et al. [27] the participants responded faster to the target when it was preceded by a regular than an irregular rhythm, even with a dual task that involved working memory. This finding supports the automatic nature of rhythm entrainment. However, in their study there was only one critical IOI–the in-phase critical IOI, and therefore we cannot differentiate between a general and a specific rhythm effect. Another study [7] included several critical IOIs, but its rhythm-induced effect was not limited to the in-phase critical IOI and it did not include an irregular rhythm condition. Therefore, it is impossible to tell whether the advantage that was observed for the in-phase critical IOI was indeed unique to this critical IOI. Specifically, Sanabria et al. [7] used a fast (IOI– 450ms) and a slow (IOI– 950ms) regular non-predictive rhythms (Experiment 3). Their participants were asked to press a key as fast as possible when hearing a target tone. Although in the fast rhythm condition the participants were indeed fastest with the in-phase critical IOI (450ms), the same critical IOI also led to the fastest RT in the slow rhythm condition even though with this condition the expected IOI was 950ms. Furthermore, by not including an irregular condition the possibility of a general effect cannot be ruled out; it is possible that the regularity of the rhythm lowered RT across all IOIs. The De la Rosa et al. [53] study is also often cited as demonstrating a specific effect of rhythm-induced attention. However, that study did not include the in-phase critical IOI (550ms), but only a multiplication of the in-phase IOI (1100ms), and a similar facilitation in comparison to an irregular rhythm was found for both this and shorter critical IOIs (i.e., both 800ms and 1100ms). Thus, although we do not doubt that an isochronous rhythm can entrain attention to a specific point in time [e.g., 1; 3; 5], such entrainment might be rather sensitive to methodological specificities.

To conclude, we found that guessing rate was lower with regular than irregular rhythms and that the quality of representation was not consistently affected by the rhythmic stimuli. These findings were limited to a block design and were found with all critical IOIs, not just the in-phase critical IOI. This lack of specificity in our study suggests that the rhythm effects found here reflect increased alertness that lowered overall signal-to-noise ratio.

## Author Contributions

**Conceptualization:** Asaf Elbaz, Yaffa Yeshurun.

**Data curation:** Asaf Elbaz.

**Formal analysis:** Asaf Elbaz.

**Funding acquisition:** Yaffa Yeshurun.

**Investigation:** Asaf Elbaz.

**Methodology:** Asaf Elbaz, Yaffa Yeshurun.

**Project administration:** Asaf Elbaz.

**Software:** Asaf Elbaz.

**Supervision:** Yaffa Yeshurun.

**Validation:** Asaf Elbaz, Yaffa Yeshurun.

**Writing – original draft:** Asaf Elbaz.

**Writing – review & editing:** Asaf Elbaz, Yaffa Yeshurun.

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
