## [Decision Letter · Decision Letter 0]

16 Oct 2019

PONE-D-19-19264

Can rhythm-induced attention improve the perceptual representation?

PLOS ONE

Dear Elbaz,

Thank you for submitting your manuscript to PLOS ONE. After careful consideration by three reviewers, we feel that it has substantial merit but does not fully meet PLOS ONE’s publication criteria as it currently stands. Therefore, we invite you to submit a revised version of the manuscript that addresses the points raised during the review process. In general, the comments seem quite addressable. Two reviewers felt some additional referencing and context would be useful in the introduction--I will leave it to your discretion to decide whether all of the suggested papers are relevant and merit inclusion.

We would appreciate receiving your revised manuscript by Nov 30 2019 11:59PM. To enhance the reproducibility of your results, we recommend that if applicable you deposit your laboratory protocols in protocols.io, where a protocol can be assigned its own identifier (DOI) such that it can be cited independently in the future. For instructions see: http://journals.plos.org/plosone/s/submission-guidelines#loc-laboratory-protocols

We look forward to receiving your revised manuscript.

Kind regards,

Jessica Adrienne Grahn

Academic Editor

PLOS ONE

Journal Requirements:

Reviewers' comments:

Reviewer's Responses to Questions

**Comments to the Author**

1. Is the manuscript technically sound, and do the data support the conclusions?

Reviewer #1: Yes

Reviewer #2: Yes

Reviewer #3: Yes

2. Has the statistical analysis been performed appropriately and rigorously? 

Reviewer #1: Yes

Reviewer #2: Yes

Reviewer #3: Yes

3. Have the authors made all data underlying the findings in their manuscript fully available?

Reviewer #1: Yes

Reviewer #2: Yes

Reviewer #3: Yes

4. Is the manuscript presented in an intelligible fashion and written in standard English?

Reviewer #1: Yes

Reviewer #2: Yes

Reviewer #3: Yes

5. Review Comments to the Author

Reviewer #1: The authors use a visual orientation task to test the role of temporal expectancies induced by a preceding rhythmic context. They also use a modelling technique to separate the contributions of guessing from the precision of the internal mental representation. In three experiments, the authors find no evidence for an enhancement for stimuli that conform to temporal expectancies. Instead, they report a more general effect of decreased guessing in a rhythmic (as opposed to irregular) context – regardless of the timing of the target stimulus. I think this paper is a useful contribution to our understanding of temporal perception, particularly through using a different methodology which provides findings that converge with recent replication failures of temporal expectation profiles. However, I also think that the contribution of the paper could be improved by relating to more of the relevant literature. The Bauer et al. (2015) paper is a good start, but there are many more papers that directly speak to this issue (e.g., Barnes & Johnston, 2010; Bermeitinger & Frings, 2015; Hickok, Farahbod, & Saberi, 2015; Kunert & Jongman, 2017; Miller, Carlson, & McAuley, 2013; Morillon, Schroeder, & Wyart, 2014; Morillon, Schroeder, Wyart, & Arnal, 2016; Prince & Sopp, 2019) in addition to a helpful review (Haegens & Zion Golumbic, 2018). It also would be good form to point the reader to the seminal papers on Dynamic Attending Theory (e.g., Jones, 1976; Jones & Boltz, 1989; Large & Jones, 1999) rather than only the 2002 article. I encourage the authors to explore this literature and incorporate material they find relevant. Otherwise I only have minor comments as detailed below.

Abstract: specify participant N for each experiment.

Line 70: “the two tasks” – did you mean “the two task types”?

Line 82: APA nitpick – “while” and “since” are temporal terms, not conjunctions

Line 94: could you clarify what exactly you mean by the height of the distribution? How is this measured? Are there standard units you could refer to? I found this measure harder to grasp than the width (given that you specified it as SD).

Line 130: “data was” – should be “were”. Also, I’m a little uncomfortable with excluding data based on model convergence failure. Is there a more empirical performance-based measure that you can use to exclude data? Same issue for Experiment 3.

Line 173: delete “a” at end of line

Line 223: this is quite a different IOI than Experiment 1, and much closer to the average spontaneous tapping rate (thus potentially making entrainment easier). Might be worth adding some justification of this design choice.

Line 396: “examined” – should be present tense

References:

Barnes, R., & Johnston, H. (2010). The role of timing deviations and target position uncertainty on temporal attending in a serial auditory pitch discrimination task. Quarterly Journal of Experimental Psychology, 63, 341-355. doi: 10.1080/17470210902925312

Bauer, A.-K.R., Jaeger, M., Thorne, J.D., Bendixen, A., & Debener, S. (2015). The auditory dynamic attending theory revisited: A closer look at the pitch comparison task. Brain Research, 1626, 198-210. doi: 10.1016/j.brainres.2015.04.032

Bermeitinger, C., & Frings, C. (2015). Rhythm and attention: Does the beat position of a visual or auditory regular pulse modulate t2 detection in the attentional blink? Frontiers in Psychology, 6. doi: 10.3389/fpsyg.2015.01847

Haegens, S., & Zion Golumbic, E. (2018). Rhythmic facilitation of sensory processing: A critical review. Neurosci Biobehav Rev, 86, 150-165. doi: 10.1016/j.neubiorev.2017.12.002

Hickok, G., Farahbod, H., & Saberi, K. (2015). The rhythm of perception: Entrainment to acoustic rhythms induces subsequent perceptual oscillation. Psychological Science, 26, 1006-1013. doi: doi:10.1177/0956797615576533

Jones, M.R. (1976). Time, our lost dimension - toward a new theory of perception, attention, and memory. Psychological Review, 83, 323-355. doi: 10.1037/0033-295X.83.5.323

Jones, M.R., & Boltz, M.G. (1989). Dynamic attending and responses to time. Psychological Review, 96, 459-491. doi: 10.1037/0033-295X.96.3.459

Kunert, R., & Jongman, S.R. (2017). Entrainment to an auditory signal: Is attention involved? Journal of Experimental Psychology: Human Perception and Performance, 146, 77-88. doi: 10.1037/xge0000246

Large, E.W., & Jones, M.R. (1999). The dynamics of attending: How people track time-varying events. Psychological Review, 106, 119-159. doi: 10.1037/0033-295X.106.1.119

Miller, J.E., Carlson, L.A., & McAuley, J.D. (2013). When what you hear influences when you see: Listening to an auditory rhythm influences the temporal allocation of visual attention. Psychological Science, 24, 11-18. doi: 10.1177/0956797612446707

Morillon, B., Schroeder, C.E., & Wyart, V. (2014). Motor contributions to the temporal precision of auditory attention. Nature Communications, 5, 5255. doi: 10.1038/ncomms6255

Morillon, B., Schroeder, C.E., Wyart, V., & Arnal, L.H. (2016). Temporal prediction in lieu of periodic stimulation. Journal of Neuroscience, 36, 2342-2347. doi: 10.1523/jneurosci.0836-15.2016

Prince, J.B., & Sopp, M. (2019). Temporal expectancies affect accuracy in standard-comparison judgements of duration, but neither pitch height, nor timbre, nor loudness. Journal of Experimental Psychology: Human Perception and Performance, 45, 585-600. doi: 10.1037/xhp0000629

Reviewer #2: In this study, the authors investigated whether entrainment to rhythms can modulate the quality of sensory representation. This is a nicely written paper with interesting results. I do not have any comments on the proposed experiments or analysis.

However, I believe it is important for authors to complement their discussion with the following points:

1) Can the task be influencing the results? As authors point out, the majority of results in entrainment use simple tasks, such as detection or discrimination tasks. Possibly, in the task authors used, the effect of entrainment can be smaller or even inexistent.

2) Whether the task can be too hard or too easy. The pattern of results suggests that participants are performing the task very well, with SD around 13 degrees. Could that be a possible reason for the lack of effects?

Reviewer #3: The current study investigated in three behavioural experiments, the influence of auditory rhythmic stimulation on visual perception. In all three experiments the authors compared the influence of a regular/irregular auditory sequence on visual target performance. While a blocked design employed in experiment 2 and 3 enhanced performance for rhythmic sequences, none of the three studies found an effect of in-phase target presentation.

Overall the methods are sound and the manuscript is well written. However, every so often there is a bit of clarification needed. Further, sometimes the motivation for the chosen parameters is not clear. Comments and suggestions are roughly listed in order of appearance within the manuscript

Major comments:

The introduction mainly focuses on attentional or neural entrainment studies in uni-modal, either auditory or visual, contexts. However, the task used in the current study is essentially a cross-modal task. It would be good to have some explicit motivation in the introduction for why an auditory rhythm should have an influence on visual target perception and in this study on visual working memory. Further, there are a few papers on cross-modal rhythms, which could be mentioned in the introduction, such as:

Miller et al., 2013: When What You Hear Influences When You See: Listening to an Auditory Rhythm Influences the Temporal Allocation of Visual Attention. Psychological Science, 24, 11-18.

Escoffier, N. et al. (2015) Auditory rhythms entrain visual processes in the human brain: Evidence from evoked oscillations and event-related potentials. Neuroimage 111, 267–276.

Barnhart, A.S. et al. (2018) Cross-modal attentional entrainment: Insights from magicians. Attention, Perception, Psychophysics. 80, 1240–1249.

p. 9, line 198ff: “Perhaps these results are due to the mixed design employed here”. This sentence is not entirely clear. First, the term mixed is a bit confounded with the mixed model used by the authors. Second, it is not mentioned in the procedure section of experiment 1 that the rhythmic and irregular trials were not presented in blocks and should be added. Further, it is not clear whether rhythmic and irregular trials were presented interleaved or whether they were randomly chosen within each block. If the latter is true, the analysis could be potentially redone on trials that were preceded by rhythmic/irregular trials (if 2 or more rhythmic trials were presented in a row), which could substantiate the claim of the authors that the results are due to the “mixed” design.

p. 11, line 255: “Instead, we found that the regular rhythm improved the quality of representation of the out of phase targets, but not that of the in-phase targets”. The authors admit that this result is a bit puzzling given the premise of attentional entrainment. As a possible explanation the authors suggest temporal expectation. First, the reasoning of their explanation is not entirely clear and should be spelled out more clearly. Second, another possible explanation might be that participants ignore input that is currently irrelevant (the rhythmic sequence) and focus solely on the critical IOIs. In that case the in-phase targets might be worse than the out of phase targets. See also:

Devergie, A., et al. (2010). Effect of rhythmic attention on the segregation of interleaved

melodies. J.Acoust.Soc.Am.128(1), EL1–EL7.

Rimmele, J., et al. (2012). Age-related changes in the use of regular patterns for auditory scene analysis. Hear.Res.289(1-2),98–107.

General discussion: as with the introduction it would be desirable that the authors discuss the results within a cross-modal context. It might be that the lack of in-phase behavioural enhancement is due to the fact that an auditory rhythm might not necessarily enhance visual perceptual performance per se.

Minor comments:

p. 7, Procedure: What were the instructions for the participants (also for experiment 2 and 3)? Were the participants instructed to pay attention to the auditory streams and thus focus on the rhythmicity or were they instructed to focus on the visual task? Manipulating the attention to either the auditory or visual modality might have an impact on the results at hand.

p. 7, line 144: which type of headphones were used?

p. 7, line 147: “(1°)”. Can the authors clarify that they mean 1° of visual angle?

p. 7, line 158: “…by a fixed IOI of 450ms.” Was there are specific reason for using 450ms? Previous studies mainly used 500ms or 600ms (Jones et al., 2002; Bauer et al., 2015; Miller et al., 2013).

p. 8, line 168: “…540 trials of which 20% were catch trials…”. It would be desirable to have the trial numbers for each critical IOI spelled out in the text. The same is true for experiment 2 and 3.

p. 9, line 215: “A total of 18 students…”. Can the authors clarify why they choose a smaller sample size as compared to experiment 1?

p. 9, line 223: “a fixed 650ms IOI”. Can the authors please clarify why they chose a different IOI and critical IOIs as compared to experiment 1? Makes it harder to compare the results from both studies.

p. 12, Procedure: it is worth mentioning in the procedure section that the trials were again presented in blocks.

p. 12, line 289: Can the authors please clarify why they chose the specific IOIs. It is not clear to me why the critical IOIs are not symmetric around the in-phase critical IOI and why the authors choose such a long critical IOI (2050ms) that is more than 2 attentional cycles from the critical IOI. What were the expectations of the authors by chosing those IOIs?

p. 13: comment: the difference in interaction results for experiment 2 and 3 might be driven by the large critical IOI of 2050ms as this IOI has a large hazard rate.

Fig. 2 irregular rhythm 250ms misses one y-axis number (0.25)

Fig. 4: it would be good if all subfigures are on the same scale (in particular the regular rhythm panels)

Fig 5: adjust the x-axis values to match the critical IOIs.

6. PLOS authors have the option to publish the peer review history of their article (what does this mean?). If published, this will include your full peer review and any attached files.

Reviewer #1: No

Reviewer #2: No

Reviewer #3: No

---

## [Author Response · Author response to Decision Letter 0]

18 Feb 2020

We would like to thank the reviewers for the time and effort they invested in our manuscript. We appreciate the comments and advices given to us and we tried to address all of them. 

Below is a detailed response to each comment made by each reviewer. Here, we would like to note two general points. First, we took this opportunity to improve the wording throughout the manuscript. Second, we found a small bug in our data processing code, and so we reanalysed all our data sets from scratch. Most critically, these reanalyses did not change the basic pattern of results reported in the original submission nor did they change the conclusions. Mostly, these reanalyses involved minor changes of the F and p values. However, it is important for us to note two changes: we realised that in the original analysis of Experiment 1 two participants were mistakenly included twice, hence, the corrected analysis of Experiment 1 includes 19 instead of 21 participants. Second, after fixing the bug, the rhythm x IOI interaction that was marginally significant for the SD parameter in Experiment 2 was no longer significant. Again this did not change the conclusions reached for this experiment in the original submission, because that marginal interaction did not follow the expected pattern of results, and therefore there and here we concluded that there are no evidence to support rhythm effect on SD.

Reviewer #1: 

The authors use a visual orientation task to test the role of temporal expectancies induced by a preceding rhythmic context. They also use a modelling technique to separate the contributions of guessing from the precision of the internal mental representation. In three experiments, the authors find no evidence for an enhancement for stimuli that conform to temporal expectancies. Instead, they report a more general effect of decreased guessing in a rhythmic (as opposed to irregular) context – regardless of the timing of the target stimulus. I think this paper is a useful contribution to our understanding of temporal perception, particularly through using a different methodology which provides findings that converge with recent replication failures of temporal expectation profiles. 

Comment: However, I also think that the contribution of the paper could be improved by relating to more of the relevant literature. The Bauer et al. (2015) paper is a good start, but there are many more papers that directly speak to this issue (e.g., Barnes & Johnston, 2010; Bermeitinger & Frings, 2015; Hickok, Farahbod, & Saberi, 2015; Kunert & Jongman, 2017; Miller, Carlson, & McAuley, 2013; Morillon, Schroeder, & Wyart, 2014; Morillon, Schroeder, Wyart, & Arnal, 2016; Prince & Sopp, 2019) in addition to a helpful review (Haegens & Zion Golumbic, 2018). It also would be good form to point the reader to the seminal papers on Dynamic Attending Theory (e.g., Jones, 1976; Jones & Boltz, 1989; Large & Jones, 1999) rather than only the 2002 article. I encourage the authors to explore this literature and incorporate material they find relevant. 

Response: We thank again the reviewer for all these suggestions. Indeed, Bauer et al. (2015) is referred to on p. 18, and we added other papers from the reviewer’s list of suggestions: Haegens & Zion Golumbic, 2018 is mentioned on p. 3, Miller et al., 2013 is mentioned on p. 6 and p. 18, and Bermeitinger & Frings, 2015; Kunert & Jongman, 2017; Prince & Sopp, 2019 are mentioned on p. 18. We also incorporated the mentioned references about the Dynamic Attending Theory (p. 3). 

Otherwise I only have minor comments as detailed below.

Comment: Abstract: specify participant N for each experiment. 

Response: Unfortunately we cannot add such specific details about the experiments to the abstract due to the limited allowed number of words  

Comment: Line 70: “the two tasks” – did you mean “the two task types”?

Response: We added “types” (p. 4).

 

Comment: Line 82: APA nitpick – “while” and “since” are temporal terms, not conjunctions

Response: Fixed.

 

Comment: Line 94: could you clarify what exactly you mean by the height of the distribution? How is this measured? Are there standard units you could refer to? I found this measure harder to grasp than the width (given that you specified it as SD).

Response: We apologize for not being clear enough. We now explain in the text that the g parameter (which is the height of the uniform distribution) reflects the proportion of trials, out of the total number of trials, in which the participant provided a random response (p. 5). Thus, if g=0.2 this means that on 20% of the trials the error is due to guessing.

Comment: Line 130: “data was” – should be “were” 

Response: Fixed. 

Comment: Also, I’m a little uncomfortable with excluding data based on model convergence failure. Is there a more empirical performance-based measure that you can use to exclude data? Same issue for Experiment 3.

Response: A model failure to ‘converge’ means that the fitting procedure could not find model’s parameters that provide good-enough fit to the data, and therefore aborted without providing parameters for the data. Because we are running the statistical analysis on these parameters, without parameters we cannot include the participants. It’s similar to excluding participants who drop from the experiment before the end – if we don’t have data for them, we cannot include them. We elaborated the explanation that was already provided in the original version to ensure it is now clear (p. 7) 

 

Comment: Line 173: delete “a” at end of line 

Response: Fixed 

Comment: Line 223: this is quite a different IOI than Experiment 1, and much closer to the average spontaneous tapping rate (thus potentially making entrainment easier). Might be worth adding some justification of this design choice.

Response: We agree and we added this to the manuscript (p. 10-11)

 

Comment: Line 396: “examined” – should be present tense

Response: Fixed  

Reviewer #2: 

In this study, the authors investigated whether entrainment to rhythms can modulate the quality of sensory representation. This is a nicely written paper with interesting results. I do not have any comments on the proposed experiments or analysis.  However, I believe it is important for authors to complement their discussion with the following points:

  Comment: 1) Can the task be influencing the results? As authors point out, the majority of results in entrainment use simple tasks, such as detection or discrimination tasks. Possibly, in the task authors used, the effect of entrainment can be smaller or even inexistent.

Response: We do not see how the task we have used could prevent the emergence of attentional entrainment to the rhythm. On the contrary, attentional effects are typically larger with more complex tasks. 

Comment: 2) Whether the task can be too hard or too easy. The pattern of results suggests that participants are performing the task very well, with SD around 13 degrees. Could that be a possible reason for the lack of effects?

Response: The reviewer is only considering here 1 type of errors – errors that reflect a non-precise encoding of the target. But there are also trials in which the participants are completely guessing. In Experiments 2 and 3 guessing rate was about 15% and 20 %, respectively. So, in addition to these trials, on trials in which they were not completely guessing (i.e., on the remaining 85% and 80% trials, respectively) they made errors with SD of about 13 deg. This leaves plenty of room for improvement due to entrainment, yet no such improvement was found.   

Reviewer #3: 

The current study investigated in three behavioural experiments, the influence of auditory rhythmic stimulation on visual perception. In all three experiments the authors compared the influence of a regular/irregular auditory sequence on visual target performance. While a blocked design employed in experiment 2 and 3 enhanced performance for rhythmic sequences, none of the three studies found an effect of in-phase target presentation.  Overall the methods are sound and the manuscript is well written. However, every so often there is a bit of clarification needed. Further, sometimes the motivation for the chosen parameters is not clear. Comments and suggestions are roughly listed in order of appearance within the manuscript  Major comments:  Comment: The introduction mainly focuses on attentional or neural entrainment studies in uni-modal, either auditory or visual, contexts. However, the task used in the current study is essentially a cross-modal task. It would be good to have some explicit motivation in the introduction for why an auditory rhythm should have an influence on visual target perception and in this study on visual working memory. Further, there are a few papers on cross-modal rhythms, which could be mentioned in the introduction, such as:  Miller et al., 2013: When What You Hear Influences When You See: Listening to an Auditory Rhythm Influences the Temporal Allocation of Visual Attention. Psychological Science, 24, 11-18. Escoffier, N. et al. (2015) Auditory rhythms entrain visual processes in the human brain: Evidence from evoked oscillations and event-related potentials. Neuroimage 111, 267–276. Barnhart, A.S. et al. (2018) Cross-modal attentional entrainment: Insights from magicians. Attention, Perception, Psychophysics. 80, 1240–1249.  Response: We added to the Introduction section a more elaborated description of the motivation for using a cross-modal design and we now refer to previous cross-modal studies (p. 5-6)

Comment: p. 9, line 198ff: “Perhaps these results are due to the mixed design employed here”. This sentence is not entirely clear. First, the term mixed is a bit confounded with the mixed model used by the authors. Second, it is not mentioned in the procedure section of experiment 1 that the rhythmic and irregular trials were not presented in blocks and should be added.

Response: We are now explicitly stating in the procedure that the regular and irregular rhythms were mixed (line 172). Additionally, we elaborated the sentence referred to by the reviewer to ensure it is not confusing (p. 9-10).

Comment: Further, it is not clear whether rhythmic and irregular trials were presented interleaved or whether they were randomly chosen within each block. If the latter is true, the analysis could be potentially redone on trials that were preceded by rhythmic/irregular trials (if 2 or more rhythmic trials were presented in a row), which could substantiate the claim of the authors that the results are due to the “mixed” design.

Response: As indicated in our response to the previous comment, we now mention in 2 different places that rhythm type varied randomly within a block. We believe it will be hard, now, to miss this point. In any case, because Experiments 2 and 3 have a blocked design they provide a stronger test of this possibility than the analysis suggested here.

Comment: p. 11, line 255: “Instead, we found that the regular rhythm improved the quality of representation of the out of phase targets, but not that of the in-phase targets”. The authors admit that this result is a bit puzzling given the premise of attentional entrainment. As a possible explanation the authors suggest temporal expectation. First, the reasoning of their explanation is not entirely clear and should be spelled out more clearly. 

Response: As we detailed above, our corrected analysis indicated that the interaction referred to in this comment, which was marginally significant in our previous analysis, is not significant after the correction. Still, the pattern of results is similar to what we presented before (i.e., no specific advantage for the in-phase point in time), and therefore we still consider a similar explanation for this pattern. Following the reviewer’s comment we elaborated the discussion of this topic and we believe it is clearer now (p. 12). 

Comment: Second, another possible explanation might be that participants ignore input that is currently irrelevant (the rhythmic sequence) and focus solely on the critical IOIs. In that case the in-phase targets might be worse than the out of phase targets. See also: Devergie, A., et al. (2010). Effect of rhythmic attention on the segregation of interleaved melodies. J.Acoust.Soc.Am.128(1), EL1–EL7. Rimmele, J., et al. (2012). Age-related changes in the use of regular patterns for auditory scene analysis. Hear.Res.289(1-2),98–107. 

Response: In Experiments 2 and 3 we did find a general effect of rhythm; this suggests that our participants did not ignore the rhythms. Additionally, because all critical IOIs had an equal probability, we don’t think we can claim that focusing on the critical IOI instead of the rhythm would benefit only the out-of-phase IOIs. Most critically, as can be seen in Fig 5b, performance is not particularly worse in the in-phase than out-of-phase conditions, only with this condition there is no ‘regular’ advantage. 

Comment: General discussion: as with the introduction it would be desirable that the authors discuss the results within a cross-modal context. It might be that the lack of in-phase behavioural enhancement is due to the fact that an auditory rhythm might not necessarily enhance visual perceptual performance per se.

Response: As we mentioned in our response to the previous comment, it is not the case that rhythm didn’t affect performance at all. We did find a general effect of rhythm. Nevertheless, we did not find an effect of rhythm that is specific to the in-phase condition. We now added a discussion of cross-modal design as a possible explanation (p. 18).   

Minor comments:  Comment: p. 7, Procedure: What were the instructions for the participants (also for experiment 2 and 3)? Were the participants instructed to pay attention to the auditory streams and thus focus on the rhythmicity or were they instructed to focus on the visual task? Manipulating the attention to either the auditory or visual modality might have an impact on the results at hand.

Response: We now clarify in the Procedure section that no instructions were given regarding the rhythms to ensure we are measuring exogenous entrainment (p. 8). 

 Comment: p. 7, line 144: which type of headphones were used?

Response: Added   

Comment: p. 7, line 147: “(1°)”. Can the authors clarify that they mean 1° of visual angle?

Response: Added 

 

Comment: p. 7, line 158: “…by a fixed IOI of 450ms.” Was there are specific reason for using 450ms? Previous studies mainly used 500ms or 600ms (Jones et al., 2002; Bauer et al., 2015; Miller et al., 2013).

Response: There was no particular reason for using 450ms, but Experiments 2 and 3 had longer IOIs and no specific effect emerged for these IOIs. Moreover, Sanabria et al., (2011) reports a specific attentional effect with 450 ms, so the particular IOIs employed do not seem to be a critical factor. 

Comment: p. 8, line 168: “…540 trials of which 20% were catch trials…”. It would be desirable to have the trial numbers for each critical IOI spelled out in the text. The same is true for experiment 2 and 3.

Response: Added.  

Comment: p. 9, line 215: “A total of 18 students…”. Can the authors clarify why they choose a smaller sample size as compared to experiment 1?

Response: We aimed at a similar number of participants in all experiments. However, it is hard to control the exact final number of participants because some have too low performance (around chance) and some have noisy data and the modelling procedure of their data cannot converge. These participants need to be excluded. Also, because some students sign up to participate in an experiment but then don’t show up, we always open few more slots for students to sign up for than what we actually want. So sometimes we end with few more participants than the aimed number. We don’t exclude participants just because we slightly passed the number we aimed for.  

Comment: p. 9, line 223: “a fixed 650ms IOI”. Can the authors please clarify why they chose a different IOI and critical IOIs as compared to experiment 1? Makes it harder to compare the results from both studies.

Response: We added an explanation for using different IOIs (p. 10-11). Basically we were trying to ensure the experimental setting is optimized for the emergence of a rhythm-induced attention allocation.

 

Comment: p. 12, Procedure: it is worth mentioning in the procedure section that the trials were again presented in blocks.

Response: Added.

  Comment: p. 12, line 289: Can the authors please clarify why they chose the specific IOIs. It is not clear to me why the critical IOIs are not symmetric around the in-phase critical IOI and why the authors choose such a long critical IOI (2050ms) that is more than 2 attentional cycles from the critical IOI. What were the expectations of the authors by chosing those IOIs?

Response: The rational for choosing these IOIs is described in details on p. 12 lines 272-281. In short, we were worried that the results of Experiment 2 were due to the fact that the in-phase IOI was also the mean IOI and that 3 critical IOIs are not enough.

  

Comment: p. 13: comment: the difference in interaction results for experiment 2 and 3 might be driven by the large critical IOI of 2050ms as this IOI has a large hazard rate.

Response: We don’t think this is the case. As can be seen when comparing Figures 5b and 7b, the difference in the patterns of results lays in the short IOIs not long ones. That is, in both experiments, with the long IOI SD was smaller in the regular than irregular condition. In contrast, with the short IOI, in Experiment 2 SD was smaller in the regular than irregular condition and vice versa for Experiment 3.

  

Comment: Fig. 2 irregular rhythm 250ms misses one y-axis number (0.25)

Response: Fixed.

  

Comment: Fig. 4: it would be good if all subfigures are on the same scale (in particular the regular rhythm panels)

Response: Fixed.

 

Comment: Fig 5: adjust the x-axis values to match the critical IOIs.

Response: Because the differences between the different IOIs are not the same it’s impossible to adjust the values of the x-axis to match all IOIs. We therefore chose the intervals between the x-axis values so that there will be a value at the most important IOI - the critical IOI (650).

---

## [Decision Letter · Decision Letter 1]

10 Mar 2020

PONE-D-19-19264R1

Can rhythm-induced attention improve the perceptual representation?

PLOS ONE

Dear Elbaz,

Thank you for submitting your manuscript to PLOS ONE. After careful consideration, we feel that it has merit but does not fully meet PLOS ONE’s publication criteria as it currently stands. Therefore, we invite you to submit a revised version of the manuscript that addresses the points raised during the review process. The reviewers felt that most of their comments were addressed, and the remaining comments are all fairly minor.

We would appreciate receiving your revised manuscript by Apr 24 2020 11:59PM. To enhance the reproducibility of your results, we recommend that if applicable you deposit your laboratory protocols in protocols.io, where a protocol can be assigned its own identifier (DOI) such that it can be cited independently in the future. For instructions see: http://journals.plos.org/plosone/s/submission-guidelines#loc-laboratory-protocols

We look forward to receiving your revised manuscript.

Kind regards,

Jessica Adrienne Grahn

Academic Editor

PLOS ONE

Reviewers' comments:

Reviewer's Responses to Questions

**Comments to the Author**

1. If the authors have adequately addressed your comments raised in a previous round of review and you feel that this manuscript is now acceptable for publication, you may indicate that here to bypass the “Comments to the Author” section, enter your conflict of interest statement in the “Confidential to Editor” section, and submit your "Accept" recommendation.

Reviewer #1: (No Response)

Reviewer #2: All comments have been addressed

Reviewer #3: (No Response)

2. Is the manuscript technically sound, and do the data support the conclusions?

Reviewer #1: Yes

Reviewer #2: Yes

Reviewer #3: Yes

3. Has the statistical analysis been performed appropriately and rigorously? 

Reviewer #1: Yes

Reviewer #2: Yes

Reviewer #3: Yes

4. Have the authors made all data underlying the findings in their manuscript fully available?

Reviewer #1: Yes

Reviewer #2: Yes

Reviewer #3: Yes

5. Is the manuscript presented in an intelligible fashion and written in standard English?

Reviewer #1: Yes

Reviewer #2: Yes

Reviewer #3: Yes

6. Review Comments to the Author

Reviewer #1: I’m happy with the revisions to the paper. I have a few minor typographical/formatting comments below.

line 143: “due to a too noisy data” – remove “a”

line 144: “analyses are conducted we had” – add a comma after “conducted”

line 176: “50m” – presumably this should be “50ms”

line 186: “trials with which” – replace “with” with “in”

line 216: “with which rhythm type” – replace “with” with “in”

page 11: be consistent with significant digits used to report p values

line 270: follow APA format for reporting F-scores below 1

line 455: “De la Rosa et al. [53] study” – add “The” before “De la Rosa”

Reference section: a number of minor APA formatting issues here, presumably these will be addressed at the copy-editing stage

Reviewer #2: (No Response)

Reviewer #3: This is the first revision of a previously submitted manuscript. I have only a few minor remarks.

p. 3, line 57 ff: “Based on such rhythm effects, the dynamic attending theory (DAT) suggests that entrainment to a rhythm reflects external rhythms driving internal oscillators leading to a narrower attentional focus with regular (isochronous) rhythms than with irregular (asynchronous) rhythms.” This sentence is somewhat confusing and would benefit from rewriting.

p. 6, line 117 ff: “For example, Miller, Carlson, and McAuley [26] have found that an auditory rhythm reduced saccade latency to a visual target and increased discrimination accuracy when target onset was in-phase with the rhythm.” Maybe rephrase to “in-phase with the preceding auditory rhythm”. In addition, it might be good to split the sentences into two sentences and mention more specifically that the visual target could occur either in-phase or out-phase.

p. 18, line 432: “Still, such attentional effects may be less robust under cross-modal setting.” It is not entirely clear what the authors would like to say with this sentence.

7. PLOS authors have the option to publish the peer review history of their article (what does this mean?). If published, this will include your full peer review and any attached files.

Reviewer #1: No

Reviewer #2: No

Reviewer #3: No

---

## [Author Response · Author response to Decision Letter 1]

17 Mar 2020

Response to reviewers 

We would like to thank the reviewers once again for the time and effort they invested in our manuscript. We were happy to learn that only minor comments were left and we addressed all of them. Below is a detailed response to each comment made by each reviewer.

Reviewer #1: 

I’m happy with the revisions to the paper. I have a few minor typographical/formatting comments below.

line 143: “due to a too noisy data” – remove “a”

line 144: “analyses are conducted we had” – add a comma after “conducted”

line 176: “50m” – presumably this should be “50ms”

line 186: “trials with which” – replace “with” with “in”

line 216: “with which rhythm type” – replace “with” with “in”

page 11: be consistent with significant digits used to report p values

line 270: follow APA format for reporting F-scores below 1

line 455: “De la Rosa et al. [53] study” – add “The” before “De la Rosa”

Reference section: a number of minor APA formatting issues here, presumably these will be addressed at the copy-editing stage

Response: We fixed all of the above comments as suggested by the reviewer, apart for the comment about the format of the references. To the best of our understanding this format should be Vancouver not APA. We therefore left the references as they are.

Reviewer #2: (No Response)

Reviewer #3: This is the first revision of a previously submitted manuscript. I have only a few minor remarks.

p. 3, line 57 ff: “Based on such rhythm effects, the dynamic attending theory (DAT) suggests that entrainment to a rhythm reflects external rhythms driving internal oscillators leading to a narrower attentional focus with regular (isochronous) rhythms than with irregular (asynchronous) rhythms.” This sentence is somewhat confusing and would benefit from rewriting.

Response: We rephrased this sentence and split it into 2 sentences.

p. 6, line 117 ff: “For example, Miller, Carlson, and McAuley [26] have found that an auditory rhythm reduced saccade latency to a visual target and increased discrimination accuracy when target onset was in-phase with the rhythm.” Maybe rephrase to “in-phase with the preceding auditory rhythm”. In addition, it might be good to split the sentences into two sentences and mention more specifically that the visual target could occur either in-phase or out-phase.

Response: Fixed

p. 18, line 432: “Still, such attentional effects may be less robust under cross-modal setting.” It is not entirely clear what the authors would like to say with this sentence.

Response: We added a sentence that clarify what we meant.

---

## [Editor Report · Decision Letter 2]

19 Mar 2020

Can rhythm-induced attention improve the perceptual representation?

PONE-D-19-19264R2

Dear Dr. Elbaz,

We are pleased to inform you that your manuscript has been judged scientifically suitable for publication and will be formally accepted for publication once it complies with all outstanding technical requirements.

With kind regards,

Jessica Adrienne Grahn

Academic Editor

PLOS ONE
---

## [Editor Report · Acceptance letter]

23 Mar 2020

PONE-D-19-19264R2 

Can rhythm-induced attention improve the perceptual representation? 

Dear Dr. Elbaz:

I am pleased to inform you that your manuscript has been deemed suitable for publication in PLOS ONE. Congratulations! Your manuscript is now with our production department. 

With kind regards,

on behalf of

Dr Jessica Adrienne Grahn 

Academic Editor

PLOS ONE